# L2 NORM GUIDED ADAPTIVE COMPUTATION

**Mani Shemiranifar**
Moallem High School
m.shemiranifar@gmail.com

**Mostafa Dehghani**
Google DeepMind
dehghani@google.com

## ABSTRACT

Although the human brain can adjust the amount of time and energy it uses to solve problems of varying complexity, many standard neural networks require a fixed computation budget regardless of the problem's complexity. This work introduces L2 Adaptive Computation (LAC), a new algorithm that adjusts the computation budget, by tracking changes in the L2 norm of a neural network's hidden state as layers are applied to the input. Unlike previous methods, LAC does not require additional trainable modules or auxiliary loss terms to make halting decisions. LAC matches the results of best-performing methods on a complex synthetic task and improves image classification accuracy while also increasing efficiency.

## 1 INTRODUCTION

The human brain relies on adaptivity in computation, enabling it to adjust the amount of time and energy dedicated to a problem based on its complexity. Conversely, standard neural networks tend to allocate the same computational budget to an input regardless of its complexity. There have been studies aimed to train models equipped with a halting mechanism that controls the computational steps (e.g., number of cell repeats in RNNs or number of layers in Transformers) based on the input's complexity and adapts the loss function to account for computational budget (Graves, 2016; Dehghani et al., 2018; Banino et al., 2021; Schuster et al., 2022; Xue et al., 2023). Following the same goal, in this work, we present L2 Adaptive Computation (LAC), a simple yet very effective method that enables a dynamic number of computation steps. Unlike most prior work, LAC does not require adding any extra module with learnable parameters to make the halting decision and requires no change on the loss function. Instead, LAC tracks changes in the L2-norm (Euclidean norm) of the model's activations as a proxy for computational progress, enabling it to determine when to halt computations. LAC is developed in Scenic (Dehghani et al., 2022) and uses JAX (Bradbury et al., 2018) and Flax (Heek et al., 2020) and the code is available at https://github.com/manishemirani/LAC.

## 2 LAC: L2 ADAPTIVE COMPUTATION

LAC requires a step function to perform, a step function can be any part of the model, e.g., a layer of the neural network. Consider $S$ as a step function where $S(x, h_t) = h_{t+1}$ where $h_0$ is initial state and $x$ is the input. We set a maximum number of computational steps T that the model can take if it does not halt before reaching step T. To make the decision whether to continue to step t+1 or halt, LAC tracks the changes in the L2 norm of the model's activations($\|h\|_2$) up to step T.

**L2 norm of model's activations** LAC offers two different ways for making the halting decision, one is per-batch halting and the other is per-example (or optionally per-tokens), where depending on the hardware and framework, either could be preferred to efficiently introduce adaptivity. With per-batch halting, the L2-norm of the activation is calculated for all examples in the batch: $\|h\|_2 = \sqrt{\Sigma_{i=1}^n \Sigma_{j=1}^d h_{ij}^2}$, where $\|h\|_2 \in \mathbb{R}^1$, where $h \in \mathbb{R}^{n \times d}$, with $d$ and $n$ denoting the hidden dimension and the batch size respectively. With per-example halting, the L2-norm of the activations is calculated separately for each individual example: $\|h\|_2 = \{\sqrt{\Sigma_{j=1}^d h_{1j}^2}, ..., \sqrt{\Sigma_{j=1}^d h_{nj}^2}\}$, where $\|h\|_2 \in \mathbb{R}^n$.

**Halting Mechanism** In LAC, the decision to halt at each step is based on the model's *progress* during that step. Specifically, we use the change in the L2-norm of the model's activation as a proxy for progress, i.e., $\delta_t = \|h_t\|_2 - \|h_{t-1}\|_2$. In other words, we stop at step $n$ if $\delta_t < \lambda_t$, where $\lambda_t$ is a

Table 1: Parity task.

| Method | Accuracy | Steps |
|---|---|---|
| RNN | 1.0 | 10 |
| RNN + `LAC` per-batch | 1.0 | 3 |
| RNN + `LAC` per-example | 1.0 | 3 |

Table 2: Image classification task.

| Method | Accuracy |
|---|---|
| ViT | 0.745 |
| UViT | 0.739 |
| ViT + `LAC` per-batch | 0.761 |
| ViT + `LAC` per-example | 0.759 |
| UViT + `LAC` per-batch | 0.743 |
| UViT + `LAC` per-example | 0.753 |

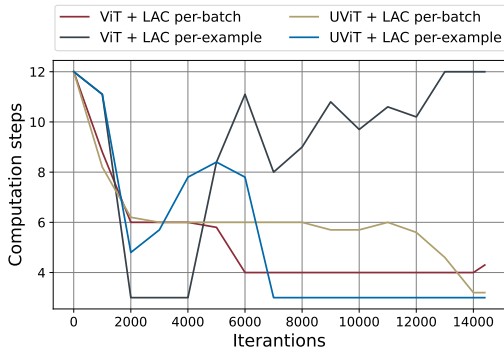

Figure 1: Number of steps taken by LAC on CI-FAR10 evaluation set with `LAC` per-example and `LAC` per-batch applied on UViT and Vit.

threshold defining the minimum activation change required for the model to be eligible to proceed to step $t + 1$. We set $\lambda_t = \alpha |\max(\Delta_t) - \min(\Delta_t)|$, where $\alpha \in (0, 1]$ is a hyperparameter that determines the decisiveness of the threshold, and $\Delta_t$ is the sequence of changes in activation across all consecutive steps before step $n$: $\Delta_t = \{\delta_{t-1}, \delta_{t-2} \ldots, \delta_1\}$. The value of $\alpha$ provides a knob to control the number of computational steps by adjusting the threshold. A lower value of $\alpha$ corresponds to a more permissive threshold, which in turn favors a larger number of computational steps.

## 3 EXPERIMENTS AND RESULTS

This section provides a condensed report of the evaluation of `LAC` on two distinct tasks, Parity and image classification. In these experiments, `LAC` was implemented to introduce adaptability to various architectures such as RNN, Vision Transformer, and Universal Transformer. Note that the value of $\alpha$ is set to 1.0 for all Parity experiments and 0.8 for image classification experiments.

**Parity**     In this section, we present the results of applying `LAC` to the parity task, in which, given a sequence of digits $\in \{1, -1, 0\}$, the model has to predict the evenness or oddness of the number of 1s in the sequence (Graves, 2016). The results of the parity task on three different architectures are presented in Table 1. RNN, RNN + `LAC` per-example, and RNN + `LAC` per-batch, where in the last two, we use a modified GRU cell to use `LAC`. All architectures have the same batch size of 128 and hidden state size of 128 and were tested on parity task with the sequence length of 16. The maximum number of computational steps, $T$, was set to 10.

**Image Classification**     We further applied `LAC` to Vision Transformer(ViT) Dosovitskiy et al. (2020) as well as Universal Vision Transformer(UViT) (Dehghani et al., 2018) (which ties the parameters of the model across layers) to evaluate `LAC` in the context of the image classification task. We use the CIFAR10 dataset, and apply no data augmentations for simplicity of comparisons. We set our ViT/UViT configuration to standard B/16 where the model has 12 layers. Table 2 shows the evaluation accuracy of `LAC` on CIFAR10, compared to the baseline. `LAC` achieved similar accuracy compared to UViT and ViT while increasing efficiency (Dehghani et al., 2021) via taking less number of computational steps (i.e., fewer layers are applied on average). Figure 1 presents the number of computational steps, in this case, encoder layers required for encoding images from the CIFAR dataset. Our approach exhibits a lower computational budget compared to UViT/ViT on the given inputs while maintaining the same level of performance.

## 4 CONCLUSION

Our paper introduces L2 Adaptive Computation (`LAC`), a new method that allows neural networks to adapt their computational budget based per patch, example, or token level. `LAC` utilizes the L2-norm of the model's activations for making halting decisions, eliminating the need for extra learnable parameters or auxiliary loss terms. Our experiments on two tasks, namely Parity and image classification, indicate that `LAC` achieves comparable results to the baselines while also improving efficiency. As an immediate future work, given the excellent cost-performance trade-off that `LAC` offers, we are seeking ways to access computational powers to apply `LAC` on large-scale training in the transfer learning setup.

## URM STATEMENT

The authors acknowledge that at least one key author of this work meets the URM criteria of the ICLR 2023 Tiny Papers Track.

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
