# OpenReview forum: "L2 Norm Guided Adaptive Computation"
_ICLR.cc/2023/TinyPapers — Submitted to Tiny Papers @ ICLR 2023_

### Official Review · Reviewer_pRkc · 2023-03-25

**Confidence:** 4

**Summary Of Contributions:**

This paper introduces L2 Adaptive Computation (LAC) that dynamically adjusts the computation budget by tracking changes in the L2 norm of a neural network’s hidden state. It eliminates the need for additional trainable modules or auxiliary loss terms to make halting decisions and shows promising results.

**Rating:**

High Potential (HP): a submission which meets the reviewing criteria and has potential to make an impact on the field

**Strengths And Weaknesses:**

Strengths:
- LAC is a simple and clear method that does not require additional training parameters and loss terms.
- The image classification results seem to be promising.
- The writing is mostly clear and easy to follow.


Weaknesses:
- For per-example and per-batch halting, the authors say "depending on the hardware and framework, either could be preferred to efficiently introduce adaptivity". Can the authors further clarify what are the pros and cons of each method and when to use which? And what is the complexity of each method?
- For the parity experiment, in table 1, the accuracies of all of the tasks are 1 even with vanilla RNN. This makes the experiments meaningless although the steps are different - cannot spot the difference before and after LAC. The authors can show the ACC of RNN with 3 steps for comparison if it's not 1.0.
- How does the performance of LAC compare with other methods that require additional parameters or losses, such as Schuster et al, 2022 and Xue et al 2023? Is it better, worse, or close?
- Clarity: (1) in section 2, the authors did not define what is $x$ in $S(x, h_n) = h_{n+1}$, (2) section 3, parity experiment, "{1or − 1, or0}", is it a typo? (3) I think the threshold can be an important hyperparameter. What is the values of it in the experiments? Are they sensitive to different experiments?

**Suggested Changes:**

- I think it's important to compare with previous methods that require additional parameters or losses, as one of the major contributions the authors claimed is simplicity. It's necessary to see the trade-off between simplicity and effectiveness.
- Maybe I missed something, but I didn't really understand Fig 1. E.g., why do the computation steps of "ViT + LAC per-example" decrease first and then increase?
- Parity experiment: see weakness.
- Can be future work: as the authors identified, it would be good to show how LAC scales to larger models/datasets.

---

> ### Author Response · Authors · 2023-05-30
> **Response to reviewer pRkc - Part 1/2**
>
> Thanks for your comments and questions. In response, we have addressed each question individually and made the necessary revisions in the paper to accommodate the comments.
>
> >  For per-example and per-batch halting, the authors say "depending on the hardware and framework, either could be preferred to efficiently introduce adaptivity". Can the authors further clarify what are the pros and cons of each method and when to use which? And what is the complexity of each method?
>
> Thank you for your question. Per-example halting refers to a network's decision-making process, where it determines whether to halt the computation for each example in a batch based on individual statistics. This allows for different examples within the batch to halt at different times. On the other hand, per-batch halting relies on aggregate statistics gathered from all examples in the batch to make the halting decisions, resulting in all examples halting at the same time. With per-batch halting, the network needs to perform computations for a number of steps equal to the maximum needed by any example in the batch.
> While per-example halting is generally preferred because it allows for more precise and granular decisions, its implementation can be computationally as expensive as per-batch halting in practice. This is because most accelerators do not perform efficiently with arbitrary sparse operations, so the computation for all examples in the batch still needs to be carried out, with only the computations for halted examples being disregarded.
>
>
>
> > For the parity experiment, in table 1, the accuracies of all of the tasks are 1 even with vanilla RNN. This makes the experiments meaningless although the steps are different - cannot spot the difference before and after LAC. The authors can show the ACC of RNN with 3 steps for comparison if it's not 1.0.
>
> We acknowledge the concern of the reviewer. While this experiment doesn't provide a comparison based solely on accuracy, it highlights the advantages of using LAC in terms of efficiency. The primary finding here is that LAC offers improved efficiency without compromising accuracy. In response to the comment about showing the accuracy of the RNN with 3 steps for comparison, we acknowledge that it could provide supplementary information. As a matter of fact, an RNN with only 3 steps also archives a perfect score in this task. However, it does not align with the main message we intend to convey in this setup. The baseline assumes a predetermined number of steps, whereas LAC does not make such an assumption. Therefore, including the RNN accuracy with 3 steps does not contribute significantly to our main point, which is adaptive computation.
>
> > How does the performance of LAC compare with other methods that require additional parameters or losses, such as Schuster et al, 2022 and Xue et al 2023? Is it better, worse, or close?
>
> The work of CALM (Schuster et al., 2022) explores early exiting with a decoder-only model, focusing on specific properties in that setup such as accounting for causal masking. On the other hand, AdaTape (Xue et al., 2023) introduces adaptivity in the input representation side rather than the architecture side. While both approaches offer control over computation adjustments, it is important to note that direct comparisons with them would not be fair or equivalent. To establish a suitable baseline, we use Universal Transformer, which incorporates ACT. In our study, we replace ACT with LAC in UT and demonstrate its superiority within the context of the Universal Transformer applied to vision tasks, specifically UViT.
>
> > Clarity: (1) in section 2, the authors did not define what is in , (2) section 3, parity experiment, "{1or − 1, or0}", is it a typo? (3) I think the threshold can be an important hyperparameter. What is the values of it in the experiments? Are they sensitive to different experiments?
>
> Thank you for your question. We have included the missing introductions to the variables in the mentioned formula. In the parity experiment, the set of digits/symbols used for the input of the parity task is `{1, -1, 0}`. For example, an input/output pair `[x, y]` could be `[-1110, 0]` or `[-1111, 1]`. We have revised the text to provide clearer clarification on this matter.  In response to your comment about the threshold value, we consistently set the threshold ($\alpha$) to $0.8$ across all image classification experiments and $1.0$ for all Parity experiments. However, it is worth noting that this value can be adjusted based on the task's difficulty. In more challenging tasks, a lower $\alpha$ value is recommended to incentivize the model to take more steps.

---

> > ### Author Response · Authors · 2023-05-30
> > **Response to reviewer pRkc - Part 2/2**
> >
> > > Maybe I missed something, but I didn't really understand Fig 1. E.g., why do the computation steps of "ViT + LAC per-example" decrease first and then increase?
> >
> > Thank you for the insightful question. We currently do not have a definitive explanation for the observed phenomenon. The ViT + LAC per-example model operates by not sharing parameters across layers and utilizes per-example statistics for halting. In this model, the delta norm  ($\delta_n$) is a vector, resulting in a different magnitude compared to per-batch halting. Our speculation is that as the model undergoes training, the halting threshold ($\lambda$), which remains a scalar across all variants, stays smaller than $\delta_n$. Additionally, since the model does not share parameters across layers, it possesses greater capacity, which further encourages it to take more steps. It is important to note that these are speculative explanations, and we are actively working to better understand the underlying mechanisms in different variants of LAC.

---

### Official Review · Reviewer_uAWN · 2023-03-26

**Confidence:** 4

**Summary Of Contributions:**

This paper proposes L2 Adaptive Computation (LAC), a mechanism to dynamically control the computational budget of deep neural networks for each input. LAC keeps track of the difference in the network's activation and decides to halt the computation when it falls below a threshold.

**Rating:**

Great Start (GS): a submission which meets some of the reviewing criteria but has room for improvement

**Strengths And Weaknesses:**

Strengths
- Adaptive computation is an interesting research topic and the authors did a fine job in introducing the problem and motivating the proposed solution
- The proposed method, LAC, is generally well-motivated, simple, and achieves encouraging results.

Weaknesses
- My major concern of this work is the unclear description of LAC. This confusion arises because the authors used $n$ to denote **both the batch size and number of computation steps**.
- The **number of computation steps** is also not well-defined for the two applications. On the image classification task, the authors stated that LAC was used to determine the number of layers to be applied. This is quite unclear as for ViT, during the forward pass, the input/feature must be forwarded until the last layer to obtain a classification prediction.
- The overall results seem interesting and encouraging, but the authors are strongly advised to clearly explain how LAC was applied on the two applications provided.

**Suggested Changes:**

Please see "Weaknesses"

---

> ### Author Response · Authors · 2023-05-30
> **Response to reviewer uAWN**
>
> We appreciate your comments and questions. We provide responses to each question. Additionally, we have incorporated the necessary changes in the paper to address the raised comments.
>
> > My major concern of this work is the unclear description of LAC. This confusion arises because the authors used
>  to denote both the batch size and number of computation steps.
>
>
>
> Thank you for your comment regarding the use of the same symbol to represent both the batch size and the number of computational steps. In sections 2 and 2.2, the symbol `n` specifically denotes the last step before LAC determines to halt the computations. In section 2.1, `n` represents the batch size of the desired hidden state. To address the overloading of this symbol, we have made revisions to the paper by replacing `n` with `t` when referring to steps, while retaining `n` to indicate the batch size.
>
>
> > The number of computation steps is also not well-defined for the two applications. On the image classification task, the authors stated that LAC was used to determine the number of layers to be applied. This is quite unclear as for ViT, during the forward pass, the input/feature must be forwarded until the last layer to obtain a classification prediction.
>
>
> We have addressed your comment and clarified the explanation in the paper. However, to provide further clarification, we define the number of computational steps as the number of times the step function is applied to inputs or features. In the parity experiment, the maximum number of computation steps was set to 10. Both LAC per example and LAC per batch consistently halted at step 3, disregarding the remaining 7 steps, across the entire evaluation set. In the image classification task, we used the B/16 configuration for Vit/UViT, which set the maximum number of computation steps (specifically, the Encoder Layers) to 12. Due to the use of Jax and Flax in our experiments, it was not feasible to halt the computation at a specific step. Instead, we pass the inputs to all 12 encoder layers, but only apply the encoder layer where LAC decides to halt. We then take the output of the last computational step (the encoder layer where LAC halts) and pass it through the remaining layers without applying further encoder layers. For example, if LAC decides to halt at step 6 (the 6th encoder layer), we use the output of this layer for the remaining 6 layers without applying additional encoder layers. It is important to note that since the shape of the layers remains the same, we can easily skip and consider the output of layer `n` where the model halted as the output of the final layer.
>
> >The overall results seem interesting and encouraging, but the authors are strongly advised to clearly explain how LAC was applied on the two applications provided.
>
>
>
> We modified the paper to provide some details on the setup for each experiment within the space limit. Additionally, we are making the code available, which contains comprehensive information and allows for result replication.

---

### Meta-Review · Area_Chair_U6Dy · 2023-04-05

**Recommendation:** Invite to archive
**Confidence:** 4

**Metareview:**

This paper is well structured and written, and promising numerical results are well presented.
It would nevertheless benefit from more carefully-designed and well-controlled experiments.

**Summary:**

L2 Adaptive Computation (LAC), a well-motivated, clear, simple, and effective trick to dynamically control the budget of NN models, is proposed.

**Reason For Not Giving A Higher Recommendation:**

- The experiments are limited in scope---more comparison with baseline using more models other than vanilla RNN would make the arguments stronger.
- The explanation on computation steps need to be clearer.

**Reason For Not Giving A Lower Recommendation:**

- The proposed method, LAC, is interesting and well-motivated.
- The paper is well written.

---

### Decision · Program_Chairs · 2023-04-08

Invite to archive